# Gene–environment pathways to cognitive intelligence and psychotic-like experiences in children

Junghoon Park[1†], Eunji Lee[2†], Gyeongcheol Cho[3], Heungsun Hwang[4], Bo-Gyeom Kim[2], Gakyung Kim[5], Yoonjung Yoonie Joo[2,6,7]*, Jiook Cha[1,2,5]*

[1]Interdisciplinary Program in Artificial Intelligence, College of Engineering, Seoul National University, Seoul, Republic of Korea; [2]Department of Psychology, College of Social Sciences, Seoul National University, Seoul, Republic of Korea; [3]Department of Psychology, College of Arts and Sciences, The Ohio State University, Columbus, United States; [4]Department of Psychology, McGill University, Montréal, Canada; [5]Department of Brain and Cognitive Sciences, College of Natural Sciences, Seoul National University, Seoul, Republic of Korea; [6]Department of Digital Health, Samsung Advanced Institute for Health Sciences & Technology (SAIHST), Sungkyunkwan University, Seoul, Republic of Korea; [7]Samsung Medical Center, Seoul, Republic of Korea

*For correspondence:
yoonjungjoo@skku.edu
(YYoonieJ);
connectome@snu.ac.kr (JC)

†These authors contributed equally to this work

Competing interest: The authors declare that no competing interests exist.

**Abstract** In children, psychotic-like experiences (PLEs) are related to risk of psychosis, schizophrenia, and other mental disorders. Maladaptive cognitive functioning, influenced by genetic and environmental factors, is hypothesized to mediate the relationship between these factors and childhood PLEs. Using large-scale longitudinal data, we tested the relationships of genetic and environmental factors (such as familial and neighborhood environment) with cognitive intelligence and their relationships with current and future PLEs in children. We leveraged large-scale multimodal data of 6,602 children from the Adolescent Brain and Cognitive Development Study. Linear mixed model and a novel structural equation modeling (SEM) method that allows estimation of both components and factors were used to estimate the joint effects of cognitive phenotypes polygenic scores (PGSs), familial and neighborhood socioeconomic status (SES), and supportive environment on NIH Toolbox cognitive intelligence and PLEs. We adjusted for ethnicity (genetically defined), schizophrenia PGS, and additionally unobserved confounders (using computational confound modeling). Our findings indicate that lower cognitive intelligence and higher PLEs are significantly associated with lower PGSs for cognitive phenotypes, lower familial SES, lower neighborhood SES, and less supportive environments. Specifically, cognitive intelligence mediates the effects of these factors on PLEs, with supportive parenting and positive school environments showing the strongest impact on reducing PLEs. This study underscores the influence of genetic and environmental factors on PLEs through their effects on cognitive intelligence. Our findings have policy implications in that improving school and family environments and promoting local economic development may enhance cognitive and mental health in children.

## eLife assessment

This study presents a **useful** inventory of the joint effects of genetic and environmental factors on psychotic-like experiences and identifies cognitive ability as a potential underlying mediating pathway. The data were analyzed using a **solid** and validated methodology based on a large, multi-center dataset. The claim that these findings are of relevance to psychosis risk and have implications for policy changes is partially supported by the results.

## Introduction

Childhood is the critical developmental period in human life. Cognitive intelligence and mental health in this period significantly impact key life outcomes at later ages, including academic performance, economic productivity, physical health, intelligence, and psychopathology (*Shonkoff, 2012*; *Walker et al., 2022*). Literature shows the significant impact of social adversities on cognitive ability and mental health in early childhood. Lower family socioeconomic status (SES), particularly household income, is linked to lower neurocognitive ability and higher risk of psychopathology in childhood (*Hair et al., 2015*; *Noble et al., 2015*; *Peverill et al., 2021*; *Tomasi and Volkow, 2021*; *Weissman et al., 2018*).

Additional to family SES, the importance of neighborhood social environment on children's neuro-cognitive ability has been also emphasized (*Gard et al., 2021*; *Tooley et al., 2020*). Adverse neighborhood environment, such as the percent of families below poverty line, low education levels, and exposure to violence, is associated with lower cognitive performance (CP) and a greater risk for psychosis in children (*Butler et al., 2018*; *Karcher et al., 2021*; *Rakesh et al., 2021*; *Taylor et al., 2020*). Conversely, as protective factors against familial and neighborhood socioeconomic challenges, supportive parenting (*Brody et al., 2017*; *Brody et al., 2019*; *Holmes et al., 2018*; *Luby et al., 2012*; *Luby et al., 2016*) and positive school environment (*Gard et al., 2021*; *Piccolo et al., 2019*; *Rakesh et al., 2021*) have been highlighted to improve child cognition and mental health.

Psychotic-like experiences (PLEs), which are prevalent in childhood, indicate the risk of psychosis (*van der Steen et al., 2019*; *van Os and Reininghaus, 2016*). Although they are not a direct precursor of schizophrenia, children reporting PLEs in ages of 9–11 years are at higher risk of psychotic disorders in adulthood (*Kelleher and Cannon, 2011*; *Poulton et al., 2000*). PLEs also point toward the potential for other psychopathologies including mood, anxiety, and substance disorders (*van der Steen et al., 2019*), are linked to deficits in cognitive intelligence (*Cannon et al., 2002*; *Kelleher and Cannon, 2011*) and show a stronger association with environmental risk factors during childhood than other internalizing/externalizing symptoms (*Karcher et al., 2021*).

Maladaptive cognitive intelligence may act as a mediator for the effects of genetic and environmental risks on the manifestation of psychotic symptoms (*Cannon et al., 2000*; *Keefe et al., 2006*; *Reichenberg et al., 2005*). Abnormal neurodevelopment, influenced by genetic factors, combined with disrupted cognitive processes resulting from socioenvironmental adversity, may eventually give rise to the positive symptoms of schizophrenia, relevant to PLEs (*Garety et al., 2001*; *Howes and Murray, 2014*). Family studies show a decline in cognitive intelligence preceding psychotic symptoms is related to genetic risk (*Cosway et al., 2000*; *Curtis et al., 2001*). In more recent studies, these associations have led to the model positing that cognitive intelligence mediates the genetic risk for psychopathology and PLEs (*Karcher et al., 2022*; *Pat et al., 2022*).

To minimize potential bias in the estimates of environmental effects, it is crucial to adjust for genetic confounding (*Sariaslan et al., 2016*), given the substantial genetic influence on intelligence (*Bouchard and McGue, 1981*; *Deary et al., 2006*; *Plomin and Spinath, 2004*) and PLEs (*Bentall and Fernyhough, 2008*; *Maxwell et al., 2023*). Recent advances in genetics have led to the development of the polygenic score (PGS) approach: a computational method to estimate the genetic loading for a complex trait using statistical associations of each single-nucleotide polymorphism (SNP) identified by genome-wide association studies (GWAS) (*Cho et al., 2022*). Particularly, PGS for two related but distinct phenotypes—CP and educational attainment (EA)—holds significant importance. As the two most frequently used proxies of cognitive intelligence in genetic studies, PGSs for CP and EA are positively correlated with intelligence, EA, income, self-rated health, and height (*Judd et al., 2020*; *Lee et al., 2018*; *Okbay et al., 2022*; *Selzam et al., 2019*). Furthermore, the PGS for EA is associated with a wide range of biological and social outcomes, including brain morphometry (*Judd et al., 2020*; *Karcher et al., 2022*), psychopathologies such as PLEs, autism, depression, Alzheimer's disease, neuroticism (*Karcher et al., 2022*; *Okbay et al., 2022*), cognitive decline (*Joo et al., 2022*; *Karcher et al., 2022*; *Ritchie et al., 2020*), body mass index (BMI), time spent watching television, geographic residence (*Abdellaoui et al., 2022*), and wealth inequality (*Barth et al., 2020*). Similar to the terms used in prior research, we will collectively refer to these two PGSs of focus as 'cognitive phenotypes PGSs' throughout this paper (*Joo et al., 2022*; *Okbay et al., 2022*; *Selzam et al., 2019*). An important gap in the literature is the lack of integrated assessment of the effects of genetic and environmental factors at multiple levels (e.g., familial vs neighborhood) to dissect the genetic and

**eLife digest** Childhood is a critical period for brain development. Difficult experiences during this developmental phase may contribute to reduced intelligence and poorer mental health later in life. Genetics and environmental factors also play roles. For example, having family support or a higher family income has been linked to better brain health outcomes for children.

Delusions or hallucinations, or other psychotic-like experiences during childhood, are linked with poor mental health later in life. Children who experience psychotic-like episodes between the ages of nine and eleven have a higher risk of developing schizophrenia or related conditions. Environmental circumstances during childhood also appear to play a crucial role in shaping the risk of schizophrenia or related conditions.

Park, Lee et al. show that positive parenting and supportive school and neighborhood environments boost child intelligence and mental health. In the experiments, Park, Lee et al. analyzed data on 6,602 children to determine how genetics and environmental factors shaped their intelligence and mental health. The models show that children with higher intelligence have a lower risk of psychosis. Both genetics and supportive environments contribute to higher intelligence.

Complex interactions between biology and social factors shape children's intelligence and mental health. Beneficial genetics and coming from a family with more financial resources are helpful. Yet, social environments, such as having parents who use positive child-rearing practices, or having supportive schools or neighborhoods, have protective effects that can offset other disadvantages. Policies that help parents, encourage supportive school environments, and strengthen neighborhoods may boost children's intelligence and mental health later in life.

environmental effects underlying abnormal cognitive intelligence and the PLEs. Addressing this with large multimodal data will allow for a more complete understanding of the factors related to the development of PLEs.

In this study, we systematically explore the longitudinal trajectories of genetic and environmental influences on PLEs, mediated through cognitive intelligence. Toward this goal, we firstly assess the associations of cognitive phenotype PGSs, family and neighborhood SES, and supportive environment with children's cognitive intelligence and longitudinally measured PLEs. To maintain robustness of our assessment, we employed statistical and computational approaches to carefully consider potential confounding. We then test the mediating effect of cognitive intelligence on the relationship between genetic and environmental factors and PLEs. Our investigation traces these effects from the baseline and through the 1- and 2-year follow-ups, providing a nuanced understanding on the role of cognitive phenotype PGSs, family SES, neighborhood SES, and positive family and school environments in shaping PLEs in children aged 9–10 years.

## Materials and methods
### Study participants
We used the multimodal genetic and environmental data of 11,878 preadolescent children aged 9–10 years old collected from 21 research sites of the Adolescent Brain Cognitive Development (ABCD) Study, one of the largest longitudinal studies for children's neurodevelopment in the United States. We analyzed the baseline, first year, and second year follow-up datasets included in ABCD Release 4.0, downloaded on January 25, 2022. After *k*-nearest neighbor imputation of missing values of covariates (categorical variables: sex, genetic ancestry, marital status of the caregiver, ABCD research sites; continuous variables: age, BMI, family history of psychiatric disorders; 4.67% of total observations imputed) using the R package VIM (*Kowarik and Templ, 2016*), we removed participants with missing data on study variables (missing genotype: $N = 3260$; follow-up observations: $N = 1180$; neighborhood information: $N = 694$; cognitive intelligence tests: $N = 126$; PLEs: $N = 5$; positive environment: $N = 11$). The final samples included 6602 multiethnic children, which comprised 890 of African ancestry (13.48%), 91 of East Asian ancestry (1.38%), 5211 of European ancestry (78.93%), 229 of Native American ancestry (3.47%), and 181 not specified (2.74%).

## Data

### NIH toolbox CP

Children's neurocognitive abilities were assessed using the NIH Toolbox Cognitive Battery, which has seven cognitive instruments for examining executive function, episodic memory, language abilities, processing speed, working memory, and attention (*Thompson et al., 2019*). We utilized baseline observations of uncorrected composite scores of fluid intelligence (Dimensional Change Card Sort Task, Flanker Test, Picture Sequence Memory Test, and List Sorting Working Memory Test), crystallized intelligence (Picture Vocabulary Task and Oral Reading Recognition Test), and total intelligence (all seven instruments) provided in the ABCD Study dataset.

### Psychotic-like experiences

Baseline and 1- and 2-year follow-up of PLEs were measured using the children's responses to the Prodromal Questionnaire-Brief Child Version. In line with previous research (*Karcher et al., 2018*; *Karcher et al., 2020*; *Karcher et al., 2021*), we computed *Total Score* and *Distress Score*, each indicating the number of psychotic symptoms and levels of total distress. Considering self- and parent-reports of psychopathology may differ (*Achenbach, 2006*), we additionally used parent-rated PLEs derived from four items of the Child Behavior Checklist according to previous studies (*Karcher et al., 2018*; *Karcher et al., 2020*; *Karcher et al., 2021*). Self- and parent-reported PLEs had significant positive correlation (Pearson's correlation of baseline year: $r = 0.095–0.0989$, $p < 0.0001$; 1-year follow-up: $r = 0.1322–0.1327$, $p < 0.0001$; 2-year follow-up: $r = 0.1569–0.1632$, $p < 0.0001$).

### Polygenic scores

To investigate the aggregated effect of genetic components, we estimated PGS of two representative cognitive phenotypes for each participant: EA and CP (*Cho et al., 2022*). We used the summary statistics released from a GWAS (*Lee et al., 2018*) of European-descent individuals for EA ($n = 1,131,881$) and CP ($n = 257,841$). EA was measured as the years of schooling; CP, measured as the respondent's score on cognitive ability assessments of general cognitive function and verbal–numerical reasoning, was assessed in participants from the COGENT consortium and the UK Biobank. To construct PGS of schizophrenia for sensitivity analyses, we used the summary statistics from the multiple GWAS of European sample ($n = 65,967$; *Ruderfer et al., 2018*) and East Asian sample ($n = 58,140$; *Lam et al., 2019*). We applied PRS-CSx, a high-dimensional Bayesian regression framework that improves cross-population prediction via continuous shrinkage prior to SNP effect sizes (*Ge et al., 2019*) (for details, see Appendix 1). The two PGSs for cognitive phenotypes had a positive significant correlation (Pearson's correlation: $r = 0.4331$, $p < 0.0001$).

### Family-, neighborhood-, and school-level environment

We assessed children's family-level SES with family income, parental education, and family's financial adversity based on parent self-reporting (*Karcher et al., 2020*; *Taylor et al., 2020*; *Tomasi and Volkow, 2021*). Higher family income and parental education and lower family's financial adversity denote a higher family SES.

Neighborhood-level SES was assessed using the *Area Deprivation Index* (ADI), the percentage of individuals below −125% of the poverty level (henceforth '*poverty*'), and *years of residence*, which were associated with PLEs in prior research (*Karcher et al., 2021*). Higher values of ADI and *poverty* and fewer *years of residence* indicate a lower neighborhood SES.

Based on existing literature (*Karcher et al., 2021*; *Rakesh et al., 2021*), we measured the level of positive parenting behavior and positive school environment to assess the effect of positive family and school environment on each individual.

### Statistical modeling

In this study, we employ linear mixed models and a novel structural equation modeling (SEM) method to examine the longitudinal trajectories of genetic and environmental influences on PLEs mediated by cognitive intelligence. We specifically investigate the mediating role of cognitive intelligence within the impacts of cognitive phenotype PGSs, high family SES, low neighborhood SES, and positive family and school environments on PLEs. These influences are examined across three periods of PLEs observations: baseline, 1-year follow-up, and 2-year follow-up (*Figure 1*).

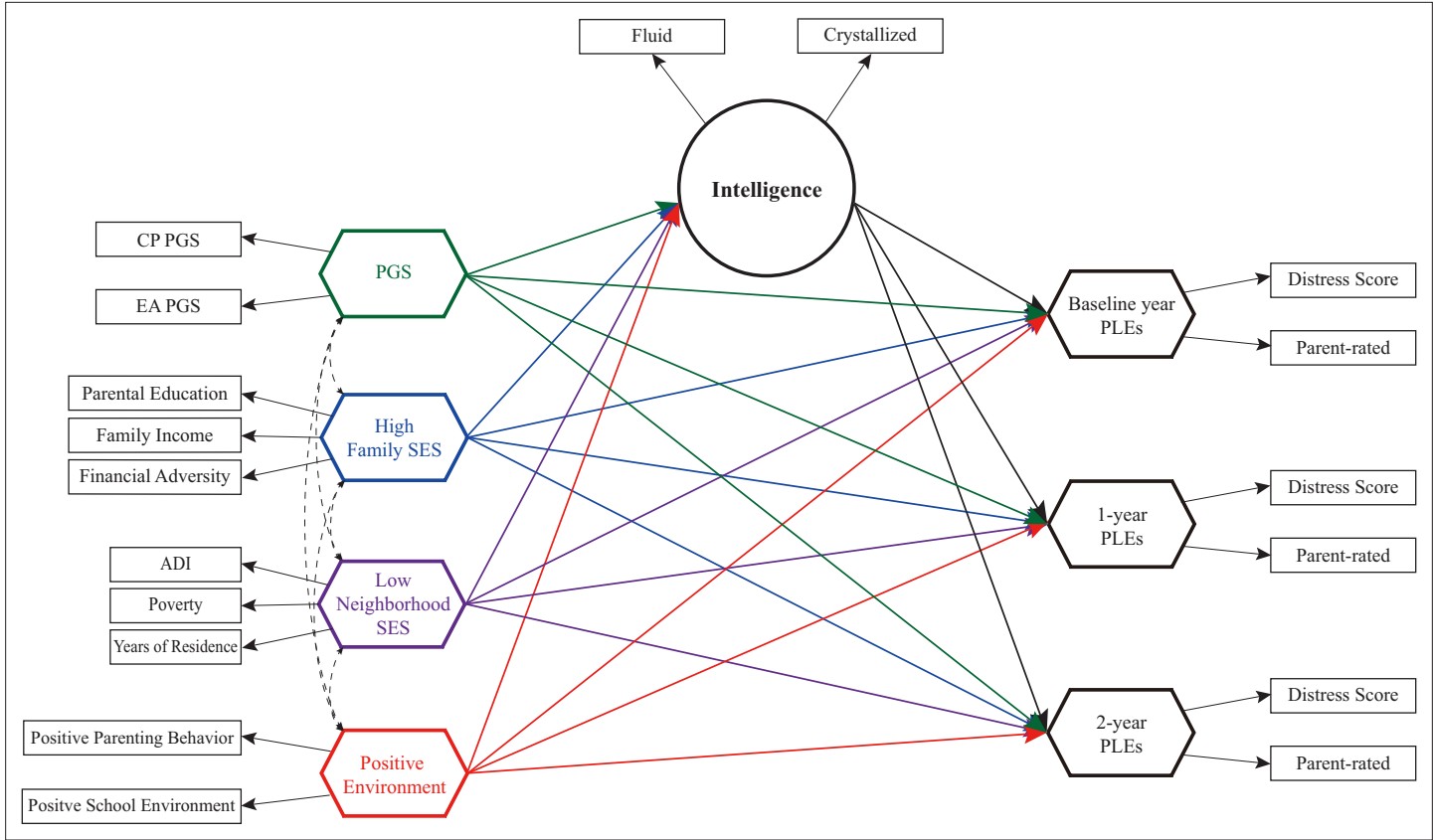

**Figure 1.** Study diagram for longitudinal trajectories of genetic and environmental influences on psychotic-like experiences (PLEs) through cognitive intelligence. This study examines the mediating role of cognitive intelligence within the effects of cognitive phenotype polygenic scores (PGSs), high family socioeconomic status (SES), low neighborhood SES, and positive family and school environments on PLEs observed at baseline, 1-year follow-up, and 2-year follow-up in children aged 9–10 years. Both direct and indirect effects, as well as total effects, were evaluated for statistical significance.

## Linear mixed models

Using linear mixed models to minimize potential population stratification from different household environments and geographical locations (*Cho et al., 2022*), we analyzed the genetic and environmental effects on cognitive intelligence and PLEs. Key variables of interest were PGS, family SES, neighborhood SES, and positive environment. To avoid multicollinearity, CP and EA PGSs were used separately in two other models. As a random factor, the variable indicating ABCD research sites was used. Variables within each model had no signs of multicollinearity (*Fox and Monette, 1992*) (generalized variance inflation factor <2.0 for every variable in all models). The child's sex, age, genetic ancestry, BMI, marital status of the caregiver, and family history of psychiatric disorders were included as covariates in the model (*Karcher et al., 2021*). All continuous variables were standardized ($z$-scaled) beforehand to get standardized estimates, and the analyses were conducted with the *lme4* package in R version 4.1.2. Throughout this paper, threshold for statistical significance was set at $p < 0.05$, with correction for multiple comparisons using the false discovery rate. 95% bootstrapped confidence intervals were obtained with 5000 iterations.

## Path modeling

To test the plausibility of whether cognitive intelligence may mediate the association between genetic and environmental factors and PLEs, we used an up-to-date SEM method, integrated generalized structured component analysis (IGSCA) (*Hwang et al., 2021b*). This approach is suited to our study using the multimodal genetic and environmental variables in that it estimates models with both factors and components as statistical proxies for the constructs.

Standard SEM using latent factors (i.e., indirectly measured indicators that explain the covariance among observed variables) to represent indicators such as PGS or family SES relies on the assumption

that observed variables within each construct share a common underlying factor. If this assumption is violated, standard SEM cannot effectively control for estimation biases. The IGSCA method addresses this limitation by allowing for the use of composite indicators (i.e., components)—defined as a weighted sum of observed variables—as constructs in the model, more effectively controlling bias in estimation compared to the standard SEM. During estimation, the IGSCA determines weights of each observed variable in such a way as to maximize the variances of all endogenous indicators and components.

We assessed path-analytic relationships among the six key constructs: cognitive phenotypes PGSs, family SES, neighborhood SES, positive family and school environment, general intelligence, and PLEs. Considering that the observed variables of the PGSs, family SES, neighborhood SES, positive family and school environment, and PLEs are evaluated as a composite index by prior research, the IGSCA method can mitigate bias more effectively by representing these constructs as components. Notably, investigations carried out by *Judd et al., 2020* and *Martin et al., 2015* utilized composite indicators to examine the genetic influence on EA and Attention-Deficit/Hyperactivity Disorder. Moreover, socioenvironmental influences are often treated as composite indicators as highlighted in *Judd et al., 2020*. When considering the psychosis continuum, studies like that of *van Os et al., 2009* postulate that psychotic disorders are likely underpinned by a multiplicity of background factors rather than a single common factor. This perspective is substantiated by a multitude of prior research that deploys composite indices for the measurement of psychotic symptoms. For these reasons, we statistically represented these constructs as the weighted sums of their observed variables or components (*Cho et al., 2022*). On the other hand, we represented general intelligence as a common factor that determines the underlying covariance pattern of fluid and crystallized intelligence, based on the classical g theory of intelligence (*Jensen, 1998*; *Spearman, 1904*).

The IGSCA model included the same covariates used in the linear mixed model as well as the ABCD research site as an additional covariate. We applied GSCA Pro 1.1 (*Hwang et al., 2021a*) to fit the IGSCA model to the data and checked the model's goodness-of-fit index (GFI) (*Jöreskog and Sörbom, 1986*), standardized root mean square residual (SRMR), and total variance of all indicators and components explained (FIT) to assess its overall goodness-of-fit. Ranging from 0 to 1, a larger FIT value indicates more variance of all variables is explained by the specified model (e.g., FIT = 0.50 denotes that the model explains 50% of the total variance of all variables) (*Hwang et al., 2021a*). The rules-of-thumb cutoff criteria in IGSCA is GFI ≥0.93 and SRMR ≤0.08 for an acceptable fit (*Cho et al., 2020*). Finally, we conducted conditional process analyses to investigate further the indirect and total effects of the constructs in the model. As a trade-off for obtaining robust nonparametric estimates without distributional assumptions for normality, the IGSCA method does not return exact p values (*Hwang et al., 2021b*). As a reasonable alternative, we obtained 95% confidence intervals based on 5000 bootstrap samples to test the statistical significance of parameter estimates.

## Sensitivity analyses

To ensure robustness of the main analyses results, we conducted multiple sensitivity analyses. As the European-descent-based GWAS was used for constructing PGS, we reran the main analyses using participants of European ancestry ($n$ = 5211) to adjust for ethnic confounding. Next, we tested effects of gene × environment interactions on cognitive intelligence and PLEs, respectively. We also tested the effects of cognitive phenotypes PGS adjusting for schizophrenia PGS, given the association of schizophrenia PGS and cognitive deficit in psychosis patients (*Shafee et al., 2018*) and individuals at-risk of psychosis (*He et al., 2021*). Lastly, we adjusted for unobserved confounding bias in the linear mixed model, using a recently developed framework for causal inference based on null treatments approach (*Miao et al., 2023*). Designed to discern causal effects from multiple treatment variables within non-randomized, observational data, the null treatments approach hinges on the assumption that no fewer than half of the confounded treatments exert no causal influence on the outcome. It circumvents the need for prior knowledge regarding which treatments are null and eliminates the necessity for independence among treatments. Given our model's inclusion of numerous treatment variables with shared variances due to the presence of unobserved confounders (*Abdellaoui et al., 2022*; *Okbay et al., 2022*)—including cognitive phenotypes PGS, family and neighborhood SES, positive family and school environments— we opted to employ this method.

**Table 1.** Demographic characteristics of the study participants.

Of the initial 11,878 ABCD samples, we obtained data for the variables of interest for 6602 multiethnic children. For multiethnic subjects (main analyses, n = 6602), 47.15% were female, and the parents of 70.21% were married. In European ancestry samples (sensitivity analyses, n = 5211), 46.71% were female, and the parents of 77.47% were married. Children of European ancestry had significantly different marital status (p < 0.0001), lower body mass index (BMI; p < 0.0001), and higher family history of psychiatric disorders (p < 0.0001) than children of multiethnic ancestries. There were no significant differences in other characteristics between the two ancestry groups. The 6602 multiethnic participants consisted of 890 African-ancestry (13.48%), 229 Native American ancestry (3.47%), 91 East Asian ancestry (1.38%), 181 not specified (2.74%), and 5211 European ancestry (78.93%) children. Differences between genetic ancestry groups were calculated using $\chi^2$ tests for categorical variables and $t$-tests for continuous variables.

| Demographic characteristics | | European ancestry (n = 5211) | | | Multiethnic (n = 6602) | | | Test statistics | |
|---|---|---|---|---|---|---|---|---|---|
| | | N | Ratio (%) | Mean (SD) | N | Ratio (%) | Mean (SD) | $t$(df)/$\chi^2$(df) | p value |
| Sex | Male | 2777 | 53.29 | | 3489 | 52.85 | | −0.4795 (11811) | 0.6316 |
| | Female | 2434 | 46.71 | | 3113 | 47.15 | | | |
| Marital status of the caregiver | Married | 4037 | 77.47 | | 4635 | 70.21 | | −10.2326 (11811) | <0.0001 |
| | Widowed | 38 | 0.73 | | 50 | 0.76 | | | |
| | Divorced | 485 | 9.31 | | 610 | 9.24 | | | |
| | Separated | 155 | 2.97 | | 232 | 3.51 | | | |
| | Never married | 275 | 5.28 | | 718 | 10.88 | | | |
| | Living with partner | 221 | 4.24 | | 357 | 5.41 | | | |
| Age (rounded to chronological month) | | | | 118.99 (7.46) | | | 118.94 (7.41) | 0.3652 (11811) | 0.715 |
| BMI | | 5211 | | 18.29 (3.67) | 6602 | | 18.72 (4.12) | −5.8889 (11811) | <0.0001 |
| Family history of psychiatric disorders (proportion of first-degree relatives who experienced mental illness) | | | | 0.10 (0.11) | | | 0.09 (0.11) | 4.4296 (11811) | <0.0001 |
| Genetic ancestry | African | - | | | 890 | 13.48 | | | |
| | Native American | - | | | 229 | 3.47 | | | |
| | East Asian | - | | | 91 | 1.38 | | | |
| | European | 5211 | 100 | | 5211 | 78.93 | | | |
| | Not specified | - | | | 181 | 2.74 | | | |

## Results

### Demographics

*Table 1* presents the demographic characteristics of the final samples. For multiethnic subjects (main analyses, n = 6602), 47.15% were female, and the parents of 70.21% were married. In European ancestry samples (sensitivity analyses, n = 5211), 46.71% were female, and the parents of 77.47% were married. Children of European ancestry showed significantly different marital status (p < 0.0001), lower BMI (p < 0.0001), and family history of psychiatric disorders (p < 0.0001) compared to children of other genetic ancestries. Our linear mixed model and IGSCA analyses were adjusted using sex, age, marital status, BMI, family history of psychiatric disorders, and ABCD research sites as covariates.

### Genetic influence on cognitive phenotypes correlates positively with intelligence and negatively with PLEs

As shown in *Figure 2*, higher PGSs of cognitive capacity phenotypes correlated significantly with higher intelligence (CP PGS: $\beta$ = 0.1113–0.1793; EA PGS: $\beta$ = 0.0699–0.1567). While CP PGS was associated only with lower baseline year *Distress Score* PLEs ($\beta$ = −0.0323), EA PGS was associated with

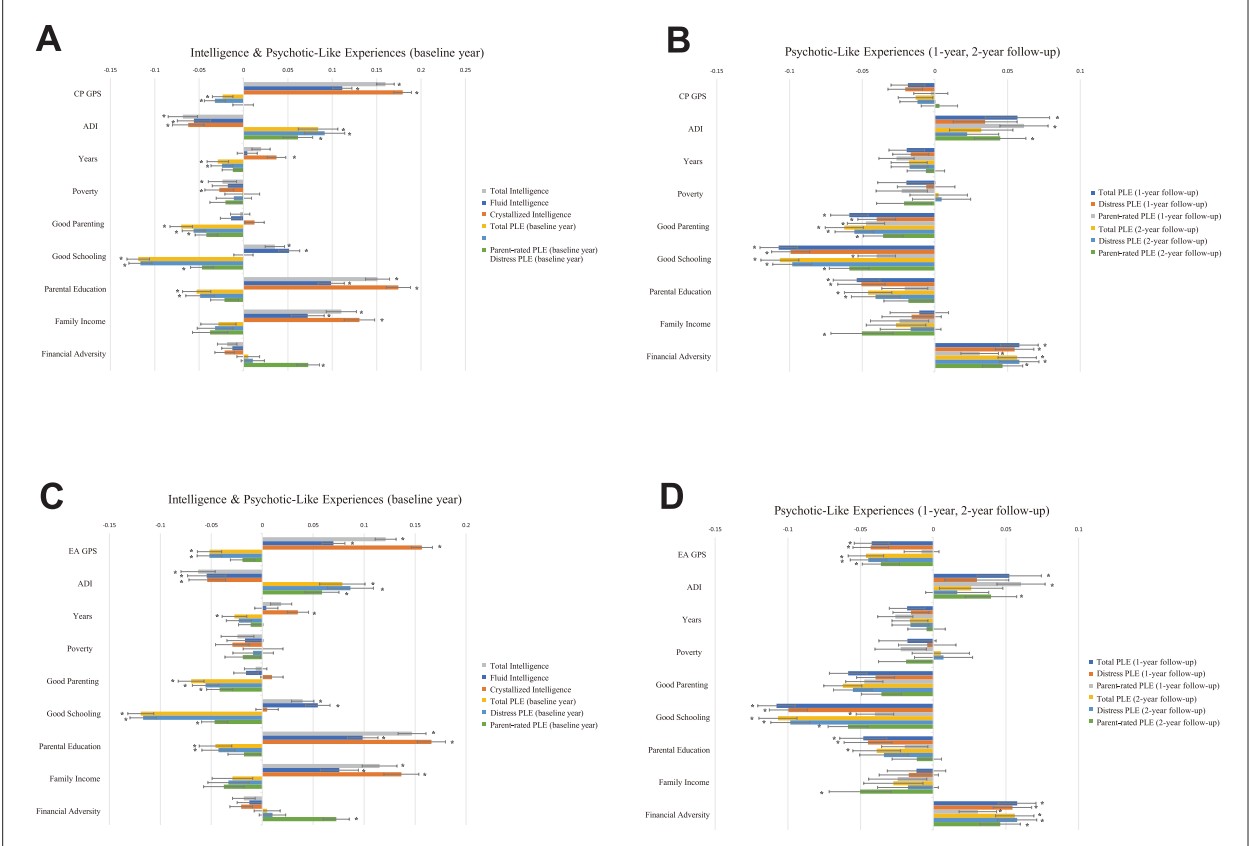

**Figure 2.** Linear models testing genetic, socioeconomic, and environmental factors associated with cognitive intelligence and PLEs. Standardized coefficients of a linear mixed model with CP PGS (**A, B**) and EA PGS (**C, D**). The analyses included 6602 samples of multiethnicity. Cognitive intelligence and PLEs correlated with the PGSs, residential disadvantage, positive environment, and family SES in the opposite directions. Error bars indicate 95% bootstrapped confidence intervals with 5000 iterations. CP and EA denote cognitive performance and education attainment, respectively; PGS, polygenic scores; SES, socioeconomic status; PLEs, psychotic-like experiences; ADI, Area Deprivation Index; Poverty, percentage of individuals below −125% of the poverty level; Years, years of residence (*p-FDR <0.05).

lower baseline year and follow-up PLEs of all measures (baseline: $\beta$ = −0.0518 to −0.0519; 1 year: $\beta$ = −0.0423 to −0.043; 2 years: $\beta$ = −0.036 to −0.0463). No significant correlations were found between CP PGS and follow-up PLEs (p > 0.05). The effects of EA PGS were larger on baseline year PLEs than follow-up PLEs. The effect sizes of EA PGS on children's PLEs were larger than those of CP PGS (EA PGS: $\beta$ = −0.036 to −0.0519; CP PGS: $\beta$ = −0.0323) (*Supplementary file 1*).

## Family and neighborhood SES correlates positively with intelligence and negatively with PLEs

Parental education associated positively with all types of intelligence ($\beta$ = 0.0699 to 0.1745) and negatively with baseline year *Total* and *Distress Score* PLEs ($\beta$ = −0.0528 to −0.043), 1-year follow-up PLEs ($\beta$ = −0.0538 to −0.0449), and 2-year follow-up PLEs ($\beta$ = −0.0459 to −0.0389). Family income correlated positively with intelligence ($\beta$ = 0.0723 to 0.1365) and negatively with 2-year follow-up parent-rated PLEs ($\beta$ = −0.0503 to −0.0502). Family's financial disadvantage correlated negatively with baseline year parent-rated PLEs ($\beta$ = 0.0726–0.0728), 1-year follow-up PLEs of all types ($\beta$ = 0.0307–0.0577), and 2-year follow-up PLEs of all types ($\beta$ = 0.0461–0.0581).

The ADI correlated significantly negatively with all types of intelligence ($\beta$ = −0.0684 to −0.054). Additionally, a higher ADI correlated significantly with higher baseline year PLEs ($\beta$ = 0.0587–0.0914), 1-year follow-up PLEs ($\beta$ = 0.0523–0.0613), and 2-year follow-up PLEs ($\beta$ = 0.0397–0.0449).

We found no significant associations of *poverty* with any of the target variables (p > 0.05). *Years of residence* correlated significantly with crystallized intelligence ($\beta$ = 0.035 to 0.0372) and baseline year *Total Score* PLEs ($\beta$ = −0.029 to −0.0273) (*Supplementary file 1*).

## Positive family and school environments correlate positively with intelligence and negatively with the influence of PLEs

Positive parenting behaviors showed significant negative correlations with baseline year PLEs ($\beta$ = −0.0702 to −0.0419), 1-year follow-up PLEs ($\beta$ = −0.0588 to −0.0397), and 2-year follow-up PLEs ($\beta$ = −0.0623 to −0.0356) (*Figure 2*). Positive school environment was associated positively with total intelligence ($\beta$ = 0.0353 to 0.0397) and fluid intelligence ($\beta$ = 0.0514 to 0.0545) and negatively with all three measures of baseline year PLEs ($\beta$ = −0.1193 to −0.0468), 1-year follow-up PLEs ($\beta$ = −0.1078 to −0.04), and 2-year follow-up PLEs ($\beta$ = −0.1068 to −0.0586) (*Supplementary file 1*).

### SEM-IGSCA

The IGSCA model showed that intelligence mediated the effects of genes and environments on the risk for psychosis (PLEs) (*Figure 3* and *Table 2*). Estimated factor loadings of latent factor variable and weights of component variables are presented in *Supplementary file 1*. Correlation matrices between component/factor variables are presented in *Appendix 3—figure 1*. The model showed a good model fit with a GFI of 0.9735, SRMR of 0.0359, and FIT value of 0.4912 (*Cho et al., 2020*). Intelligence was under significant direct influences of the cognitive phenotypes PGS ($\beta$ = 0.2427), family SES ($\beta$ = 0.2932), neighborhood SES ($\beta$ = −0.1121), and positive environment ($\beta$ = 0.0268). Family SES and positive environment had significant negative direct effects on PLEs of all years. Cognitive phenotypes PGS and neighborhood SES showed no significant direct effects on any of the PLEs (p > 0.05). Intelligence significantly mediated the effects of the PGS, family and neighborhood SES, and positive environment on PLEs of all years: the PGS (baseline year: $\beta$ = −0.035; 1 year: $\beta$ = −0.0355; 2 years: $\beta$ = −0.0274), family SES (baseline year: $\beta$ = −0.0423; 1 year: $\beta$ = −0.0429; 2 years: $\beta$ = −0.0331), neighborhood SES (baseline year: $\beta$ = 0.0162; 1 year: $\beta$ = 0.0164; 2 years: $\beta$ = 0.0126), and positive environment (baseline year: $\beta$ = −0.0039; 1 year: $\beta$ = −0.0039; 2 years: $\beta$ = −0.003).

For all observed years, positive environment had largest total effects on PLEs (baseline year: $\beta$ = −0.152; 1 year: $\beta$ = −0.1316; 2 years: $\beta$ = −0.1364), followed by family SES (baseline year: $\beta$ = −0.1216; 1 year: $\beta$ = −0.1119; 2 years: $\beta$ = −0.1164), neighborhood SES (baseline year: $\beta$ = 0.0504; 1 year: $\beta$ = 0.0329; 2 years: $\beta$ = 0.0192), and PGS (baseline year: $\beta$ = −0.0498; 1 year: $\beta$ = −0.036; 2 years: $\beta$ = −0.0365). The total effects of each indicator on PLEs were significant except for those of neighborhood SES (p > 0.05).

### Sensitivity analyses

As sensitivity analyses, we reran our main analyses with adjustment for ethnic confounding, schizophrenia PGS, and unobserved confounding, respectively. Results of linear mixed model and IGSCA analyses were consistent (*Supplementary file 1*). See Appendix 2 for detailed results.

## Discussion

This study investigated the relationships of the genetic and environmental influences on the development of intelligence and the PLEs in youth, leveraging genetic data from the large epidemiological samples and a multi-level environmental (socioeconomic) data. Our results support the model that genetic factors (PGS for cognitive phenotypes), socioeconomic conditions, and family and school environments may influence cognitive intelligence in children, and this impact may lead to the individual variability of the current and future PLEs in children. Our analysis with integrated data shows the contributions of genetic and environmental factors, respectively, to cognitive and mental wellness in children, and further provides policy implications to improve them.

Our SEM shows that cognitive intelligence mediates the environmental and genetic influence on the current and future PLEs. The environmental factors (family SES, neighborhood SES, and positive parenting and schooling) and PGS of cognitive phenotypes exhibit significant indirect effects via cognitive intelligence on PLEs. Prior research identifying the mediation of cognitive intelligence focused on either genetic (*Karcher et al., 2022*) or environmental factors (*Lewis et al., 2020*) alone. Studies with older clinical samples have shown that cognitive deficit may be a precursor for the onset of psychotic disorders (*Eastvold et al., 2007*; *Fett et al., 2020*; *Vorstman et al., 2015*). Our study advances this by demonstrating the integrated effects of genetic and environmental factors on PLEs through the cognitive intelligence in 9- to 11-year-old children. Such comprehensive analysis contributes to

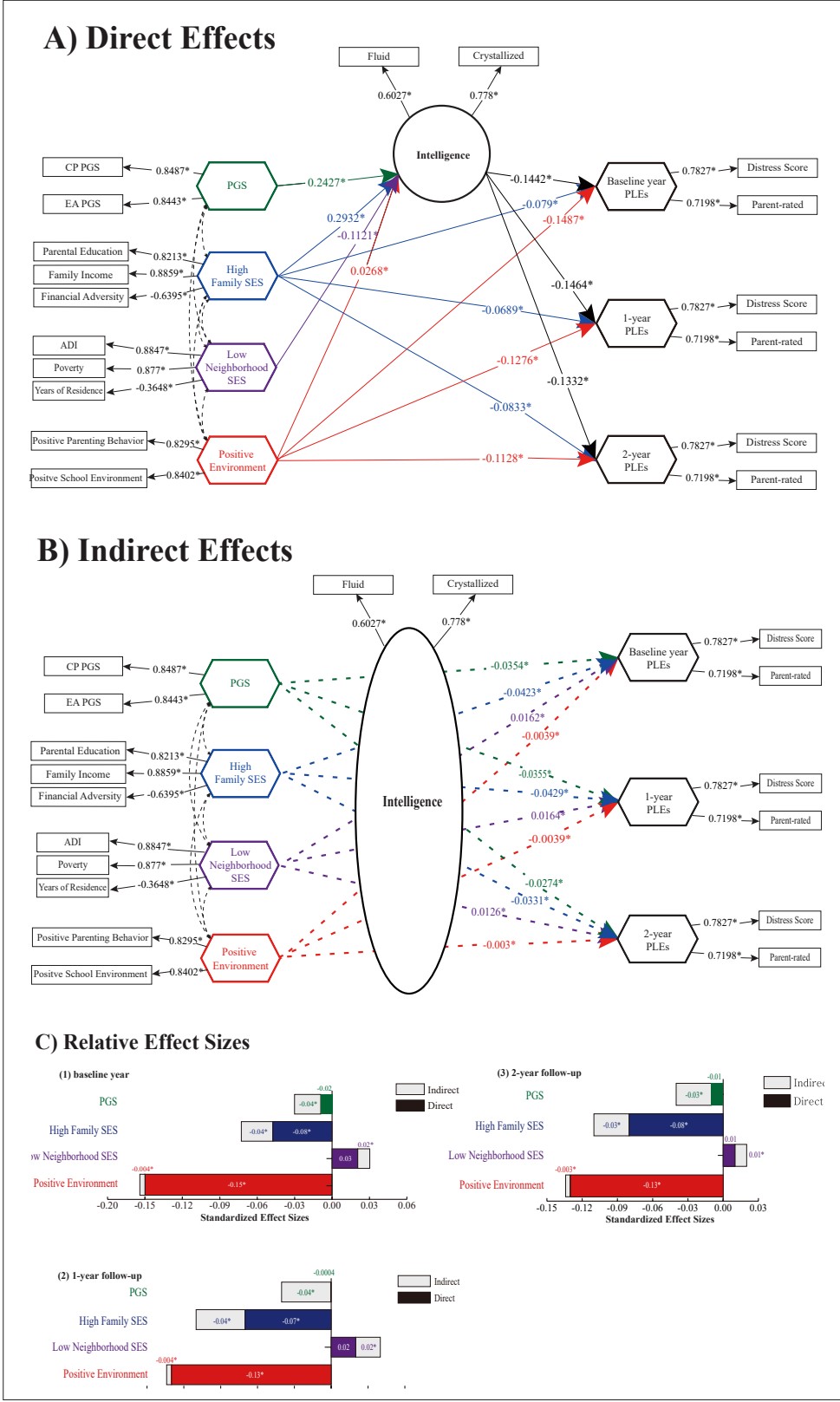

**Figure 3.** Direct/indirect effects of gene–environment factors to cognitive and PLEs outcomes. (**A**) Direct pathways from PGS, high family SES, low neighborhood SES, and positive environment to cognitive intelligence and PLEs. Standardized path coefficients are indicated on each path as direct effect estimates (significance level *p < 0.05). (**B**) Indirect pathways to PLEs via intelligence were significant for polygenic scores, high family SES, low

*Figure 3 continued on next page*

*Figure 3 continued*

neighborhood SES, and positive environment, indicating the significant mediating role of intelligence. (**C**) Relative effect sizes of direct and indirect pathways within the total effects on PLEs. The standardized effect sizes of direct pathways are colored within each bar. In (**A–B**), child sex, genetic ancestry, body mass index (BMI), marital status, family history of psychiatric disorders, and ABCD research sites were included as covariates. CP PGS and EA PGS denote polygenic scores of cognitive performance and education attainment, respectively; SES, socioeconomic status; PLEs, psychotic-like experiences; Crystallized and Fluid, crystallized and fluid intelligence; ADI, Area Deprivation Index; Poverty, percentage of individuals below −125% of the poverty level; Years, years of residence. Note: * indicates a statistically significant parameter estimate at α = 0.05.

assessing the relative importance of various factors influencing children's cognition and mental health, and it can aid future studies designed for identifying health policy implications. Considering the directions and magnitudes of the effects, though the effects of PGS remain significant, aggregated effects of environmental factors account for much greater degrees on PLEs.

Our results of cognitive intelligence mediating the genetic and environmental effects on PLEs may be related to several potential mechanisms. Children raised in higher family SES may have sufficient nutrition and cognitive stimulants, whereas children living in deprived neighborhoods may be exposed to higher rates of crime, air pollution, and substance abuse (*Lewis et al., 2020*; *Marshall et al., 2020*; *Taylor et al., 2020*; *Tomasi and Volkow, 2021*). Environmental enrichment may be associated with longer periods of neural plasticity (e.g., myelination, maturation of brain circuitry), leading to higher cognitive ability and lower risk of mental disorders like PLEs (*Tooley et al., 2021*). This may be further linked to the cognitive reserve theory. The theory suggests that genetic influence for cognitive phenotypes and environmental enrichment promotes more efficient, flexible brain networks, which may lead to greater resilience against psychopathology (*Stern, 2009*). Indeed, prior clinical studies show the linkage between cognitive reserve and psychosis (*Amoretti et al., 2018*; *Leeson et al., 2011*).

Our results indicate that genetic influences on cognitive phenotypes are significantly linked to PLEs. PGSs for CP and EA were strongly correlated with PLEs (baseline year, 1-year follow-up, and 2-year follow-up). These associations were robust after adjustment for schizophrenia PGS, ethnic confounding and unobserved confounders. Cognitive phenotypes PGS generally show higher predictive performance than PGS of any other traits (*Lee et al., 2018*; *Okbay et al., 2022*; *Plomin and von Stumm, 2018*). Genetic variants associated with CP and EA are related to complex traits across the life span, including neuroticism, depressive symptoms, smoking in adulthood, cognitive decline at a later age (*Joo et al., 2022*), risk for Alzheimer's disease (*Lee et al., 2018*; *Okbay et al., 2022*), brain volume, area, and thickness, as well as psychotic disorders (*Karcher et al., 2022*). Prior gene expression studies suggest that polygenic signals for schizophrenia, bipolar disorder, and EA are significantly enriched in the central nervous system, particularly the cerebellum (*Finucane et al., 2018*). Our findings emphasize the importance of cognitive phenotypes PGS as a biomarker which not only implicates cognitive traits but also exhibits genetic overlap with the PLEs.

The differing magnitudes of the PGS impact between EA and CP warrant attention. The effects of the EA PGS on the PLEs of all years were 160.68–371.67% larger than those of CP PGS. This discrepancy may result from that the larger sample size of EA GWAS than that of CP GWAS. Alternatively, the discrepancies in effect sizes may suggest different genetic compositions between EA and CP. Recent literature documents that more than half of the polygenic signal for EA is related to noncognitive and social skills required for successful EA (*Demange et al., 2021*), whereas CP may rather be linked to cognitive skills. This observation also supports the well-established relationships of the EA PGS with socioeconomic and life-course outcomes (e.g., social mobility *Belsky et al., 2018*), voter turnout (*Aarøe et al., 2021*), BMI, income, time spent watching television, geographic residence (*Abdellaoui et al., 2022*), and wealth inequality (*Barth et al., 2020*), which may be influenced by unobserved environmental factors (*Young et al., 2019*). In our analysis, the utilization of two PGSs a more comprehensive evaluation, contributing to an estimation of the genetic and environmental factors that attempted to minimize confounding bias.

Furthermore, the significant effects of cognitive phenotypes PGS on cognitive intelligence ($\beta$ = 0.0699–0.1793) remained robust and similar in magnitude after adjusting for genetic ancestry ($\beta$ = 0.0754–0.1866) and other (unobserved) confounding ($\beta$ = 0.0546–0.1776). As we controlled for family-, neighborhood-, and school-level environmental factors and unobserved confounders, our results may

**Table 2.** Integrated generalized structured component analysis (IGSCA) of multiethnic samples.

Sex, age, genetic ancestry, body mass index (BMI), parental education, marital status of the caregiver, household income, and family's financial adversity based on parents' self-report, family history of psychiatric disorders, and ABCD research sites were included as covariates. Family socioeconomic status was included to confirm that the aassociations of polygenic score (PGS), neighborhood disadvantage, and positive environment are meaningful. SE and CI represent standard error and confidence intervals, respectively. Significant effects are marked with a star (*).

Analysis of total/direct/indirect effects

| Effect type | Paths | Estimate | SE | 95% CI | | Significance |
|---|---|---|---|---|---|---|
| **Effects from PGS to intelligence (baseline year)** | | | | | | |
| Direct effect | PGS → intelligence | 0.242736 | 0.01277 | 0.218202 | 0.267954 | * |
| **Effects from high family SES to intelligence (baseline year)** | | | | | | |
| Direct effect | High family SES → intelligence | 0.293171 | 0.016737 | 0.260337 | 0.326413 | * |
| **Effects from low neighborhood SES to intelligence (baseline year)** | | | | | | |
| Direct effect | Low neighborhood SES → intelligence | −0.1121 | 0.016768 | −0.14568 | −0.08118 | * |
| **Effects from positive environment to intelligence (baseline year)** | | | | | | |
| Direct effect | Positive environment → intelligence | 0.026793 | 0.012552 | 0.003984 | 0.052633 | * |
| **Effects from intelligence to psychotic-like experiences (all years)** | | | | | | |
| Direct effect | Intelligence → psychotic-like experiences (baseline year) | −0.14421 | 0.027683 | −0.20344 | −0.09516 | * |
| Direct effect | Intelligence → psychotic-like experiences (1-year follow-up) | −0.14638 | 0.027507 | −0.20834 | −0.09983 | * |
| Direct effect | Intelligence → psychotic-like experiences (2-year follow-up) | −0.11276 | 0.028708 | −0.17428 | −0.063 | * |
| **Effects from PGS to psychotic-like experiences (baseline year)** | | | | | | |
| Total effect | PGS → psychotic-like experiences | −0.05017 | 0.011354 | −0.07292 | −0.02853 | * |
| Indirect effect | PGS → intelligence → psychotic-like experiences | −0.035 | 0.007126 | −0.0508 | −0.02273 | * |
| Direct effect | PGS → psychotic-like experiences | −0.01516 | 0.01347 | −0.04085 | 0.012389 | |
| **7 Effects from high family SES to psychotic-like experiences (baseline year)** | | | | | | |
| Total effect | High family SES → psychotic-like experiences | −0.12126 | 0.019087 | −0.15851 | −0.08313 | * |
| Indirect effect | High family SES → intelligence → psychotic-like experiences | −0.04228 | 0.008652 | −0.06139 | −0.02707 | * |
| Direct effect | High family SES → psychotic-like experiences | −0.07898 | 0.020747 | −0.11856 | −0.03698 | * |
| **Effects from low neighborhood SES to psychotic-like experiences (baseline year)** | | | | | | |
| Total effect | Low neighborhood SES → psychotic-like experiences | 0.050374 | 0.018277 | 0.013545 | 0.085934 | * |
| Indirect effect | Low neighborhood SES → intelligence → psychotic-like experiences | 0.016166 | 0.003944 | 0.009843 | 0.025298 | * |
| Direct effect | Low neighborhood SES → psychotic-like experiences | 0.034209 | 0.0184 | −0.00268 | 0.069813 | |
| **Effects from positive environment to psychotic-like experiences (baseline year)** | | | | | | |
| Total effect | Positive environment → psychotic-like experiences | −0.15256 | 0.013871 | −0.17965 | −0.1252 | * |
| Indirect effect | Positive environment → intelligence → psychotic-like experiences | −0.00386 | 0.002065 | −0.00859 | −0.00058 | * |
| Direct effect | Positive environment → psychotic-like experiences | −0.14869 | 0.014025 | −0.17573 | −0.12073 | * |
| **Effects from PGS to psychotic-like experiences (1-year follow-up)** | | | | | | |
| Total effect | PGS → psychotic-like experiences | −0.035895 | 0.011646 | −0.058499 | −0.013458 | * |
| Indirect effect | PGS → intelligence → psychotic-like experiences | −0.03553 | 0.007062 | −0.05176 | −0.02376 | * |
| Direct effect | PGS → psychotic-like experiences | −0.00036 | 0.013579 | −0.02566 | 0.028107 | |

*Table 2 continued on next page*

*Table 2 continued*

Analysis of total/direct/indirect effects

| | | | | | | |
|---|---|---|---|---|---|---|
| **Effects from high family SES to psychotic-like experiences (1-year follow-up)** | | | | | | |
| Total effect | High family SES → psychotic-like experiences | −0.11184 | 0.018291 | −0.1478 | −0.07584 | * |
| Indirect effect | High family SES → intelligence → psychotic-like experiences | −0.04291 | 0.008569 | −0.06242 | −0.0288 | * |
| Direct effect | High family SES → psychotic-like experiences | −0.06892 | 0.019586 | −0.10522 | −0.02866 | * |
| **Effects from low neighborhood SES to psychotic-like experiences (1-year follow-up)** | | | | | | |
| Total effect | Low neighborhood SES → psychotic-like experiences | 0.032947 | 0.018055 | −0.00264 | 0.068773 | |
| Indirect effect | Low neighborhood SES → intelligence → psychotic-like experiences | 0.016409 | 0.004003 | 0.010133 | 0.025893 | * |
| Direct effect | Low neighborhood SES → psychotic-like experiences | 0.016538 | 0.018503 | −0.02066 | 0.051855 | |
| **Effects from positive environment to psychotic-like experiences (1-year follow-up)** | | | | | | |
| Total effect | Positive environment → psychotic-like experiences | −0.13149 | 0.013154 | −0.15756 | −0.10589 | * |
| Indirect effect | Positive environment → intelligence → psychotic-like experiences | −0.00392 | 0.00208 | −0.0087 | −0.00059 | * |
| Direct effect | Positive environment → psychotic-like experiences | −0.12757 | 0.013237 | −0.15343 | −0.10137 | * |
| **Effects from PGS to psychotic-like experiences (2-year follow-up)** | | | | | | |
| Total effect | PGS → psychotic-like experiences | −0.03643 | 0.012196 | −0.06027 | −0.01272 | * |
| Indirect effect | PGS → intelligence → psychotic-like experiences | −0.02737 | 0.007142 | −0.04307 | −0.01508 | * |
| Direct effect | PGS → psychotic-like experiences | −0.00906 | 0.014737 | −0.03696 | 0.021152 | |
| **Effects from high family SES to psychotic-like experiences (2-year follow-up)** | | | | | | |
| Total effect | High family SES → psychotic-like experiences | −0.11632 | 0.018067 | −0.15258 | −0.08174 | * |
| Indirect effect | High family SES → intelligence → psychotic-like experiences | −0.03306 | 0.008796 | −0.05228 | −0.01807 | * |
| Direct effect | High family SES → psychotic-like experiences | −0.08326 | 0.019462 | −0.12066 | −0.04392 | * |
| **Effects from low neighborhood SES to psychotic-like experiences (2-year follow-up)** | | | | | | |
| Total effect | Low neighborhood SES → psychotic-like experiences | 0.01921 | 0.018684 | −0.01767 | 0.055261 | |
| Indirect effect | Low neighborhood SES → intelligence → psychotic-like experiences | 0.012641 | 0.003814 | 0.006533 | 0.0215 | * |
| Direct effect | Low neighborhood SES → psychotic-like experiences | 0.006569 | 0.019176 | −0.03173 | 0.042823 | |
| **Effects from positive environment to psychotic-like experiences (2-year follow-up)** | | | | | | |
| Total effect | Positive environment → psychotic-like experiences | −0.13627 | 0.013881 | −0.1635 | −0.10926 | * |
| Indirect effect | Positive environment → intelligence → psychotic-like experiences | −0.00302 | 0.001703 | −0.0069 | −0.00043 | * |
| Direct effect | Positive environment → psychotic-like experiences | −0.13325 | 0.014009 | −0.16069 | −0.10565 | * |

be interpreted as significant genetic influences on individual's cognitive intelligence. This interpretation is supported by a recent study (*Isungset et al., 2022*): Despite of the socioeconomic differences in Norway (a typical social democratic welfare state) and the United States (a typical liberal welfare state), the magnitudes of genetic influence on cognitive intelligence were similar (Norway: $\beta$ = 0.18; United States: $\beta$ = 0.17). Cognitive phenotypes PGS is an important genetic factor across the nations and societies. Therefore, analyses omitting the genetic influence may be subject to overestimation of the socioeconomic impact (*Plomin and von Stumm, 2018*; *Sariaslan et al., 2016*).

This study shows that a high SES and positive environment, particularly positive parenting behavior and school environment, is associated with higher intelligence and a lower risk for PLEs in children. While prior research has emphasized the dominant role of family SES (e.g., family income) (*Tomasi and Volkow, 2021*), our SEM analyses (IGSCA) showed that positive environmental factors such as supportive parenting and schooling have a greater impact on children's PLEs. Specifically, the effect sizes were the highest in supportive family and school environment, followed by family and neighborhood SES. Even after adjusting for genetic ancestry and unobserved confounders, the strong associations of positive parenting and schooling with higher intelligence and fewer PLEs remained significant. These findings suggest that interventions that target positive family and school environments may be particularly effective. Recent research supports this notion, showing that interventions that promote supportive parenting and inclusive school environments can improve neurocognitive ability, academic performance, and decrease risk behaviors such as drinking and emotional eating (*Brody et al., 2017*; *Brody et al., 2019*; *Holmes et al., 2018*).

Moreover, our results showed that positive parenting and schooling in baseline year were associated not only with baseline year PLEs but also with PLEs 1–2 years later. This is in line with prior research showing that intervention focused on parenting behavior and school environment have long-lasting positive effects that extend into adulthood and even across generations (*Cunha and Heckman, 2007*; *Hill et al., 2020*).

While policy implications in observational studies like ours might be limited, our findings show the importance of comprehensive approaches considering the entire ecosystem of children's lives—including residential, family, and school environment—for future research aimed to enhance children's intelligence and mental health. When we combine the total effect sizes of neighborhood and family SES, as well as positive school environment and parenting behavior $\left(\sum |\beta| = 0.2718 \sim 0.3242\right)$, they considerably surpass the total effect sizes of cognitive phenotypes PGSs $\left(|\beta| = 0.0359 \sim 0.0502\right)$. It has been suggested that a holistic and quantitative approach that takes into account the comprehensive ecosystem of family, school, and residential environments may ensure policy effects and efficient use of resources (*Cree et al., 2018*; *Garner et al., 2021*; *Shonkoff, 2012*). For example, the Health Impact in 5 Years Initiative of the US Centers for Disease Control and Prevention (*CDC, 2018*) includes 14 evidence-based interventions, such as providing school-based prevention programs, public transportation, home improvement loans, and earned income tax credits, to tackle the social determinants of public health. Our study strengthens the idea that an interdisciplinary science-driven, coordinated approach to intervening in the select environmental factors may allow practical improvements in child development, particularly in those who are at a disadvantage.

Our study has some limitations. First, due to data availability constraints in the ABCD Study, we only utilized baseline observations for NIH Toolbox cognitive intelligence, and we could not test whether PLEs might be a mediator of intelligence. Second, the generalizability of our findings may be limited since most of the participants included in our analysis are from European ancestry. Although the ABCD Study aimed to achieve its representativeness by recruiting from an array of school systems located around each of the 21 research sites, chosen for their diversity in geography, demographics, and SES, it is not fully representative of the US population (*Compton et al., 2019*). Third, the duration of the follow-up period utilized in this study is relatively short (1- and 2-year follow-up), which may limit the interpretability of our findings for understanding cognitive and psychiatric development during later childhood. Future research could potentially benefit from employing longer follow-up periods, as more follow-up observations are being collected in the ABCD Study. Fourth, while we used a wide range of statistical methods to adjust for confounding bias from observed and unobserved variables (e.g., genetic ancestry), we did not account for other types of potential bias such as sample selection bias. Fifth, despite a number of causal inference methods used in this study, the ABCD Study is a non-randomized dataset. Given the observational nature of the ABCD Study, interpreting our results as actual causality requires more caution. Finally, we did not include all important environmental variables, such as air pollution (*Marshall et al., 2020*) and social capital (*Krabbendam and van Os, 2005*), which are not collected in the ABCD Study.

In conclusion, our study provides potential pathways of genetic factors of cognitive phenotypes and environmental factors of family, school, and neighborhood to cognitive and mental wellness in children. Our findings underscore the importance of a comprehensive approach that considers both

biological and socioeconomic features in promoting young children's cognitive ability and mental health. Given the importance of child development, it requires joint efforts from multiple disciplines.

## Acknowledgements

This work was supported by the National Research Foundation of Korea (NRF) grant funded by the Korea government (MSIT) (No. 2021R1C1C1006503, 2021K1A3A1A2103751212, 2021M3E5D2A01022515, RS-2023-00266787, RS-2023-00265406, 021R1I1A1A01054995, and RS-2023-00250759), by Creative-Pioneering Researchers Program through Seoul National University (No. 200-20230058), and by Institute of Information & communications Technology Planning & Evaluation (IITP) grant funded by the Korea government (MSIT) [No. 2021-0-01343, Artificial Intelligence Graduate School Program (Seoul National University)], and the New Faculty Startup Fund from Sungkyunkwan University (S-2023-1896-000-01).

## Additional information

### Funding

| Funder | Grant reference number | Author |
| --- | --- | --- |
| National Research Foundation of Korea | 2021R1C1C1006503 | Jiook Cha |
| National Research Foundation of Korea | 2021K1A3A1A2103751212 | Jiook Cha |
| National Research Foundation of Korea | 2021M3E5D2A01022515 | Jiook Cha |
| National Research Foundation of Korea | RS-2023-00266787 | Jiook Cha |
| National Research Foundation of Korea | RS-2023-00265406 | Jiook Cha |
| Seoul National University | Creative-Pioneering Researchers Program through Seoul National University 200-20230058 | Jiook Cha |
| Seoul National University | Semi-Supervised Learning Research Grant by SAMSUNG A0426-20220118 | Jiook Cha |
| Institute for Information and Communications Technology Planning and Evaluation | NO.2021-0-01343 | Jiook Cha |
| Institute for Information and Communications Technology Planning and Evaluation | Artificial Intelligence Graduate School Program | Jiook Cha |

The funders had no role in study design, data collection, and interpretation, or the decision to submit the work for publication.

### Author contributions

Junghoon Park, Conceptualization, Validation, Investigation, Methodology, Writing - original draft, Project administration, Writing – review and editing; Eunji Lee, Investigation, Writing - original draft, Writing – review and editing; Gyeongcheol Cho, Methodology; Heungsun Hwang, Investigation, Methodology; Bo-Gyeom Kim, Gakyung Kim, Investigation; Yoonjung Yoonie Joo, Supervision, Validation, Investigation, Writing – review and editing; Jiook Cha, Conceptualization, Supervision, Funding acquisition, Writing – review and editing

## Author ORCIDs

Junghoon Park (ID) https://orcid.org/0000-0001-8982-0387
Eunji Lee (ID) http://orcid.org/0000-0002-2589-5395
Gyeongcheol Cho (ID) https://orcid.org/0000-0002-9237-0388
Heungsun Hwang (ID) http://orcid.org/0000-0002-5057-7479
Bo-Gyeom Kim (ID) http://orcid.org/0000-0003-2685-877X
Yoonjung Yoonie Joo (ID) http://orcid.org/0000-0001-9506-8742
Jiook Cha (ID) https://orcid.org/0000-0002-5314-7992

Joint Public Review: https://doi.org/10.7554/eLife.88117.4.sa1
Author response https://doi.org/10.7554/eLife.88117.4.sa2

## Additional files

### Supplementary files

• MDAR checklist

• Supplementary file 1. Supplementary file containing linear mixed model analyses results, factor loadings of integrated generalized structured component analysis, and sensitivity analyses results.

• Source data 1. Synthetic dataset for replication (not used in actual analysis).

### Data availability

All codes used in this study can be found at https://github.com/Transconnectome/Intell_PLE_Pathway (copy archived at *Park, 2024*). The original ABCD Study dataset is freely accessible to all qualified researchers upon submission of an access request through the National Institutes of Mental Health Data Archive (nda.nih.gov). Each of the participating centers secured comprehensive written informed consent from parents and assent from all children involved. The research protocols received approval from the University of California, San Diego's Institutional Review Board (IRB) under approval number 160091, as well as from the IRBs of the 21 data collection sites involved (*Auchter et al., 2018*). Due to ABCD Study's policy in data sharing, we provide a synthetic dataset instead of real observations. This synthetic dataset was generated with conditional GAN (*Xu et al., 2019*) to imitate the data structure of our final study samples. After automatic hyperparameter optimization with Optuna (*Akiba et al., 2019*), the synthetic dataset showed overall quality score of 84.15%. Note that analyses results from the synthetic data may not be 100% identical to the results presented in this paper, due to the differences between synthetic vs original dataset.

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

## Appendix 1

### Genotype data

The ABCD Study collected saliva samples of study participants at the baseline visit and shipped the samples from the collection site to the Rutgers University Cell and DNA Repository. Genomic DNA was extracted and genotyped using the Affymetrix NIDA Smokescreen array (733,293 SNPs). We removed any inferior SNPs with (1) genotype call rate <95%, (2) sample call rate <95%, and (3) rare variants with minor allele frequency (MAF) <0.01. Variant imputation was performed with the Michigan Imputation Server (*Das et al., 2016*) using the 1000 Genome phase 3 version 5 multiethnic Grch37/hg19 reference panel with Eagle ver2.4 phased output (*Loh et al., 2016*). For the imputed 12,046,090 SNPs, we only retained data from any individuals with <5% missing genotypes; without extreme heterozygosity (*F* coefficient <3 standard deviations from the population mean); and SNPs with >0.4 imputation quality INFO score, <5% missingness rate, >0.01 MAF and Hardy–Weinberg equilibrium (p > $10^{-6}$). The genetic ancestry of each participant was determined with the fastSTRUCTURE algorithm (*Raj et al., 2014*), available from ABCD Release 4.0. Considering the diverse ethnic and racial backgrounds of the study participants, we estimated both kinship coefficients (K.C.s) and ancestrally informative principal components (P.C.s) to additionally control familial relatedness and ancestry admixture using PC-Air (*Conomos et al., 2015*) and PC-Relate (*Conomos et al., 2016*). We selected unrelated participants that were inferred to be more distant than fourth-degree relatives (K.C. >0.022) and removed any outliers that fell significantly outside (>6 S.D. limits) the center in P.C. space. In the rest of this paper, we used final genotype data (11,301,999 variants) of 10,199 unrelated multiethnic samples, including 7893 European-ancestry participants, after Q.C.

### Polygenic scores

Hyperparameters of PGSs were optimized in the held-out validation set of 1579 unrelated participants, consisting of 88 of African ancestry (5.57%), 25 of East Asian ancestry (1.58%), 1365 of European ancestry (86.45%), 88 of Native American ancestry (2.91%), and 55 not specified (3.48%). The validation set was created during the quality control process of the study genotype data when we selected unrelated study participants of the original ABCD samples with pairwise kinship coefficients of less than 0.022 among them. To select the optimal hyperparameter, we fitted linear regression models for intelligence composite scores within the validation set and evaluated the model performance in terms of the highest $R^2$ and effect size (beta). The models were adjusted for age, sex, and the first 10 principal components of genotype data. For the PGS of multiethnic participants, genetic ancestry determined by ADMIXTURE (*Alexander et al., 2009*) was additionally included as a covariate. The validation samples were only used for hyperparameter tuning for PGS optimization and were excluded from any further analyses. The PGSs were residualized against the first 10 ancestrally informative P.C.s to adjust for population stratification.

### Parent-rated psychotic-like experiences

As self- and parent-reports of psychopathology often differ, parent-rated PLEs derived from four items of the Child Behavior Checklist:

1. 'Hears sounds or voices that other people think aren't there.'
2. 'Sees things that other people think aren't there.'
3. 'Does things that other people think are strange.'
4. 'Has thoughts that other people would think are strange.'

Each question was scored from 0 = not true, 1 = somewhat or sometimes true, and 2 = very true or often true.

### Family and neighborhood SES

We assessed children's family SES based on family income, parental education, and family's financial adversity. All three variables were based on self-reported responses of children's primary caregiver. For family income, the caregivers were asked '*What is your TOTAL COMBINED FAMILY INCOME for the past 12 months? This should include income (before taxes and deductions) from all sources, wages, rent from properties, social security, disability and/or veteran's benefits, unemployment benefits, workman's compensation, help from relative (include child payments and alimony), and so on. If Separated/Divorced, please average the two household incomes.*' Answer choices are shown below:

1. Less than $5000
2. $5000 through $11,999
3. $12,000 through $15,999
4. $16,000 through $24,999
5. $25,000 through $34,999
6. $35,000 through $49,999
7. $50,000 through $74,999
8. $75,000 through $99,999
9. $100,000 through $199,999
10. $200,000 and greater

Parental education was measured as the highest grade or level of school completed or highest degree received by the primary caregiver. ('What is the highest grade or level of school you have completed or the highest degree you have received?'). Answer choices were:

1. Never attended/Kindergarten only
2. 1st grade
3. 2nd grade
4. 3rd grade
5. 4th grade
6. 5th grade
7. 6th grade
8. 7th grade
9. 8th grade
10. 9th grade
11. 10th grade
12. 11th grade
13. 12th grade
14. High school graduate
15. GED or equivalent diploma
16. Some college
17. Associate degree: Occupational
18. Associate degree: Academic Program
19. Bachelor's degree (ex. BA)
20. Master's degree (ex. MA)
21. Professional School degree (ex. MD)
22. Doctoral degree (ex. PhD)

Family's financial adversity is measured with Parent-Reported Financial Adversity Questionnaire, reflecting family's financial ability to pay for basic life expenses (*Diemer et al., 2013*). The questionnaire asked whether the child's caregiver and family experienced any of the following difficulties within the past 12 months:

1. 'Needed food but couldn't afford to buy it or couldn't afford to go out to get it?'
2. 'Were without telephone service because you could not afford it?'
3. 'Didn't pay the full amount of the rent or mortgage because you could not afford it?'
4. 'Were evicted from your home for not paying the rent or mortgage?'
5. 'Had services turned off by the gas or electric company, or the oil company wouldn't deliver oil because payments were not made?'
6. 'Had someone who needed to see a doctor or go to the hospital but didn't go because you could not afford it?'
7. 'Had someone who needed a dentist but couldn't go because you could not afford it?'

Each of the seven items was scored with 0=no or 1=yes.

Neighborhood SES was measured by Residential History Derived Scores based on the census tracts of each respondent's primary address. The national percentile score of ADI (*Karcher et al., 2021*; *Rakesh et al., 2021*) was calculated from the 2011–2015 American Community Survey 5-year summary. Components used for deriving ADI includes:

1. Percentage of population aged ≥25 years with <9 years of education
2. Percentage of population aged ≥25 years with at least a high school diploma
3. Percentage of employed persons aged ≥16 years in white collar occupations
4. Median family income
5. Income disparity defined by Singh as the log of 100x ratio of the number of households with <10,000 annual income to the number of households with >50,000 annual income.
6. Median home value
7. Median gross rent
8. Median monthly mortgage
9. Percentage of owner
10. Percentage of occupied housing units with >1 person per room (crowding)
11. Percentage of civilian labor force population aged ≥16 years unemployed (unemployment rate)
12. Percentage of families below the poverty level
13. Percentage of population below 138% of the poverty threshold
14. Percentage of single
15. Percentage of occupied housing units without a motor vehicle
16. Percentage of occupied housing units without a telephone
17. Percentage of occupied housing units without complete plumbing (log)

## Positive family and school environment

Positive parenting behavior was assessed using the ABCD Children's Report of Parental Behavioral Inventory. We used the average values of mean summary scores of five questionnaire items about first and second caregivers. Single values were used for respondents lacking a second caregiver. Positive school environment was assessed as the sum of children's responses to 12 items in the ABCD School Risk and Protective Factors Survey.

Prior work used response items about the first caregiver for positive parenting behavior and the first six items about the school environment for positive school environment (*Rakesh et al., 2021*). However, to obtain a comprehensive and accurate assessment, we also included items about the second caregiver and six additional questions about the school environment.

Children were asked to choose between 1 = Not like him/her, 2 = Somewhat like him/her, or 3 = A lot like him/her about the five questions regarding the first and second caregivers:

1. Smiles at me very often.
2. Is able to make me feel better when I am upset.
3. Believes in showing his/her love for me.
4. Is easy to talk to.
5. Makes me feel better after talking over my worries with him/her

The following questions were asked to assess positive school environment:

1. In my school, students have lots of chances to help decide things like class activities and rules.
2. I get along with my teachers.
3. My teacher(s) notices when I am doing a good job and lets me know about it.
4. There are lots of chances for students in my school to get involved in sports, clubs, or other school activities outside of class.
5. I feel safe at my school.
6. The school lets my parents know when I have done something well.
7. I like school because I do well in class.
8. I feel I'm just as smart as other kids my age.
9. There are lots of chances to be part of class discussions or activities.
10. In general, I like school a lot.
11. Usually, school bores me.
12. Getting good grades is not so important to me.

Each of the 12 items was scored with 1 = NO!, 2 = no, 3 = yes, 4 = YES!. Item 11 and 12 were reverse coded when obtaining summary scores.

## Appendix 2

### Results of linear mixed models with European samples

As the European-descent-based GWAS was used for constructing PGS, we reran the main analyses using participants of European ancestry (*n* = 5211) to adjust for ethnic confounding (females 46.71%, mean age 118.99 [SD 7.46]).

The results of linear mixed models were similar to those of main analyses (*Supplementary file 1*). Higher intelligence was significantly associated with higher CP and EA PGS (*β*s > 0.0754, p < 0.0001), lower ADI (*β*s < −0.0503, p = 0.0253), more *years of residence* (*β*s > 0.0268, p = 0.0311), positive school environment (*β*s > 0.0365, p = 0.004), higher parental education (*β*s > 0.1067, p < 0.0001), more family income (*β*s > 0.0561, p = 0.0166), and less family's financial disadvantages (*β*s < −0.0427, p = 0.011). No significant association of *poverty* and supportive parenting behavior with intelligence was found (p > 0.05).

More Total and Distress Score PLEs were significantly associated with lower CP and EA PGS (baseline: *β*s < −0.0316, p = 0.0212; 1 year: *β*s < −0.0422, p = 0.0016; 2 years: *β*s < −0.0489, p = 0.0002), higher ADI (baseline: *β*s > 0.0554, p = 0.0421), less supportive parenting (baseline: *β*s < −0.0722, p < 0.0001; 1 year: *β*s < −0.0473, p = 0.0013; 2 years: *β*s < −0.0693, p < 0.0001), less positive school environment (baseline: *β*s < −0.1076, p < 0.0001; 1 year: *β*s < −0.0896, p = 0.0013; 2 years: *β*s < −0.0847, p < 0.0001), lower parental education (baseline: *β*s < −0.0632, p = 0.0011; 1 year: *β*s < −0.0567, p = 0.004; 2 years: *β*s < −0.0463, *P*=0.0198), more family's financial disadvantage (1 year: *β*s > 0.0605, p = 0.005; 2 years: *β*s > 0.0646, p < 0.0001). No significant association of *years of residence*, *poverty*, and family income with Total and Distress Score PLEs was found (p > 0.05). Parent-rated PLEs was positively associated with ADI (baseline: *β* = 0.0486, p = 0.0378; 1 year: *β*s > 0.0527, p = 0.0225) and negatively with positive school environment (baseline: *β*s < −0.0522, p = 0.0018; 2 years: *β*s < −0.0601, p = 0.0009) and family's financial disadvantage (2 years: *β*s > 0.0501, p = 0.0216).

### Results of IGSCA with European samples

The results of IGSCA in European ancestry samples were similar to those in multiethnic participants (*Supplementary file 1*). It showed a good model fit with a GFI of 0.9695, SRMR of 0.0397, and FIT value of 0.4854.

Intelligence was under a significant direct influence of the cognitive phenotypes PGS (*β* = 0.2987 [95% CI = 0.2673 to 0.3281]), neighborhood SES (*β* = −0.0931 [95% CI = −0.1303 to −0.0586]), family SES (*β* = 0.3034 [95% CI = 0.2683 to 0.3413]), and positive environment (*β* = 0.0396 [95% CI = 0.0104 to 0.0698]). Neighborhood SES (*β* = 0.0411 [95% CI = 0.0037 to 0.0784]), family SES (*β* = −0.0531 [95% CI = −0.0956 to −0.0101]), and positive environment (*β* = −0.1473 [95% CI = −0.1779 to −0.1163]) showed significant direct influence on baseline year PLEs, but cognitive phenotypes PGS did not. Constructs that had significant direct effects on PLEs of 1- and 2-year follow-up were family SES (*β*s < −0.0850) and positive environment (*β*s < −0.1222).

Intelligence had a significant mediating effect on the pathways of the PGS (*β* = −0.0555 [95% CI = −0.0754 to −0.0392]), family SES (*β* = −0.0563 [95% CI = −0.07749 to −0.0392]), neighborhood SES (*β* = 0.0173 [95% CI = 0.0098 to 0.0271]), and positive environment (*β* = −0.0074 [95% CI = −0.0142 to −0.0019]) to baseline year PLEs. For PLEs of 1- and 2-year follow-up, the mediation effects of intelligence were also significant with all four constructs: PGS (1 year: *β* = −0.0397 [95% CI = −0.0586 to −0.0235]; 2 years: *β* = −0.0204 [95% CI = −0.0389 to −0.0036]), family SES (1 year: *β* = −0.0404 [95% CI = −0.0601 to −0.0238]; 2 years: *β* = −0.0207 [95% CI = −0.0399 to −0.0036]), neighborhood SES (1 year: *β* = 0.0124 [95% CI = 0.0064 to 0.0208]; 2 years: *β* = 0.0064 [95% CI = 0.0011 to 0.0132]), and positive environment (1 year: *β* = −0.0053 [95% CI = −0.0108 to −0.0012]; 2 years: *β* = −0.0027 [95% CI = −0.0067 to −0.0002]). Positive environment had the largest total effects on PLEs among the constructs (baseline year: *β* = −0.1547; 1 year: *β* = −0.1274; 2 years: *β* = −0.1386).

### Results of linear mixed models adjusted for schizophrenia PGS with multiethnic samples

We also assessed whether the effects of cognitive phenotypes PGS in the linear mixed model are significant including schizophrenia PGS. The results showed that inclusion of schizophrenia PGS in the models did not change much of the main results (*Supplementary file 1*). Schizophrenia PGS

was negatively associated with total and fluid intelligence in EA PGS model (Total intelligence $\beta$ = −0.0293, 95% CI = −0.0488 to −0.01, p = 0.0048; Fluid intelligence $\beta$ = −0.0317, 95% CI = −0.0522 to −0.0097, p = 0.0062). No significant association was found between schizophrenia PGS and PLEs of all three time points (p > 0.05).

## Results of linear mixed models adjusted for unobserved confounders with multiethnic samples

Unobserved confounding variables may bias linear regression estimates. In particular, linear estimates may be subject to collider bias when those unobserved confounders and genetic factors are jointly associated with environmental factors and target traits (*Akimova et al., 2021*). Thus, bias from unobserved confounding variables were adjusted using the null treatments approach (*Miao et al., 2023*). This approach can identify the causal effects of multiple treatment variables in presence of unobserved confounders. Assuming that fewer than half of the treatments have causal effects on the outcome, it first estimates the joint distribution of treatments–confounders to obtain asymptotic bias using a factor model. Second, it estimates the relationship between treatments, confounders, and outcome using standard density estimation methods (e.g., least squares regression). Finally, by eliminating the asymptotic bias from the estimated treatments–confounders–outcome relationship, it can adjust for unobserved confounding even without specifying which treatments are null.

The significance of effects of PGSs was mostly preserved after adjusting for unobserved confounders (*Supplementary file 1*). Higher cognitive phenotypes PGSs correlated significantly with higher intelligence (CP PGS: $\beta$s > 0.1111, p < 0.0001; EA PGS: $\beta$s > 0.0546, p = 0.0002). CP PGS was associated with lower baseline year Distress Score PLEs ($\beta$ = −0.0348, p = 0.021) and EA PGS was associated with lower Total and Distress Score PLEs of baseline year and follow-up years (baseline: $\beta$s < −0.0534, p = 0.0002; 1 year: $\beta$ = −0.0375, p = 0.0149; 2 years: $\beta$s < −0.0391, p = 0.017). We found no significant associations of neighborhood SES variables with any of the target variables (p > 0.05).

While positive parenting behaviors did not have significant association with intelligence, positive school environment was significantly associated with higher total and fluid intelligence ($\beta$s > 0.0328, p = 0.0242). Positive parenting behaviors had significant negative correlations with all types of PLEs of baseline year and follow-up years (baseline: $\beta$s < −0.0412, p = 0.036; 1 year: $\beta$s < −0.0435, p = 0.0316; 2 years: $\beta$s < −0.0558, p = 0.0017). Positive school environment was also negatively associated with all types of PLEs of baseline year and follow-up years (baseline: $\beta$s < −0.0464, p = 0.036; 1 year: $\beta$s < −0.0952, p < 0.0001; 2 years: $\beta$s < −0.0583, p = 0.0095).

Parental education and family income were not significantly associated with all types of intelligence and PLEs of all three time points. Family's financial disadvantage did not show significant association with intelligence and baseline year PLEs, but it had positive associations with 1-year follow-up Total Score ($\beta$ = 0.0604, p = 0.0113) and Distress Score PLEs ($\beta$ = 0.0532, p = 0.0197) and 2-year follow-up Total Score ($\beta$ > 0.0544, p = 0.0284) and Distress Score PLEs ($\beta$ = 0.0584, p = 0.0224).

## Appendix 3

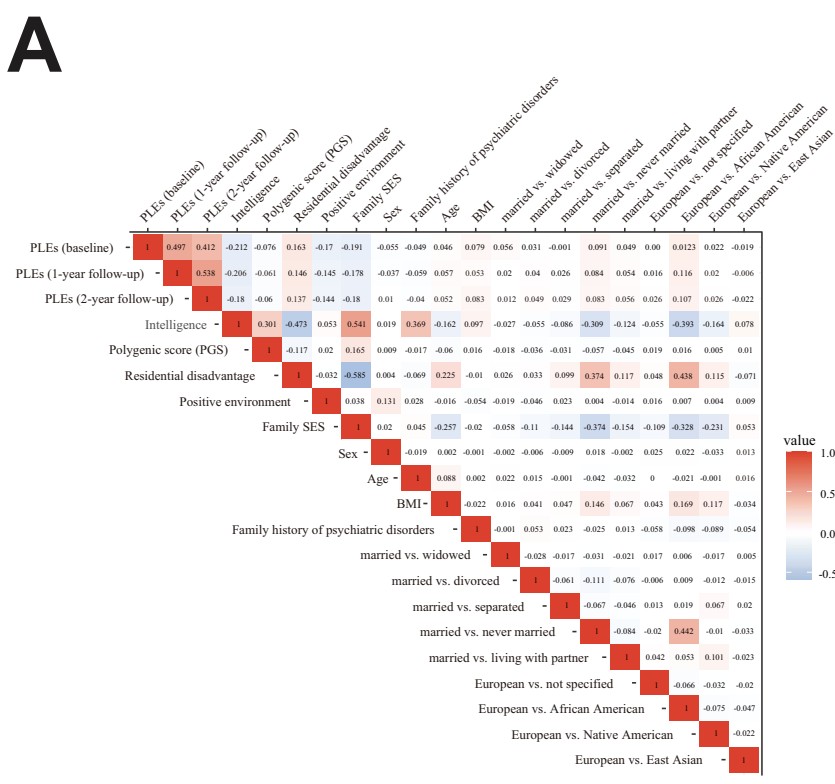

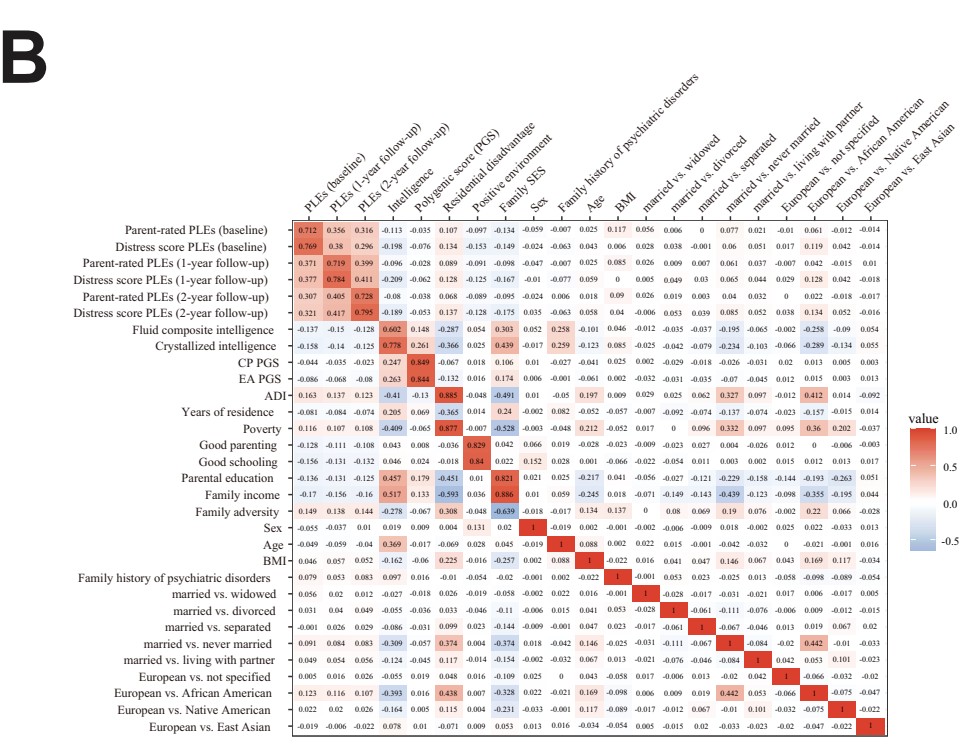

**Appendix 3—figure 1.** Component correlation matrix of integrated generalized structured component analysis (IGSCA). (**A**) Correlation between all component/factor variables of the IGSCA model. (**B**) Correlation between all observed variables used to construct the relevant component/factor variables in the IGSCA model. CP PGS and EA PGS denote polygenic scores of cognitive performance and education attainment, respectively; PLEs, psychotic-like experiences; Crystallized and Fluid, crystallized and fluid intelligence; ADI, Area Deprivation Index; Poverty, percentage of individuals below −125% of the poverty level; Years, years of residence.

