## [Editor Report · eLife assessment]

This study presents a **useful** inventory of the joint effects of genetic and environmental factors on psychotic-like experiences and identifies cognitive ability as a potential underlying mediating pathway. The data were analyzed using a **solid** and validated methodology based on a large, multi-center dataset. The claim that these findings are of relevance to psychosis risk and have implications for policy changes is partially supported by the results.

---

## [Referee Report · Joint Public Review]

This paper aimed to assess the link between genetic and environmental factors on psychotic-like experiences and the potential mediation through cognitive ability. This study was based on data from the ABCD cohort, including 6,602 children aged 9-10 years. The authors report a mediating effect, suggesting that cognitive ability is a key mediating pathway in linking several genetic and environmental (risk and protective) factors to psychotic-like experiences.

Strengths of the methods: The authors use a wide range of validated (genetic, self- and parent-reported, and cognitive) measures in a large dataset with a 2-year follow-up period. The statistical methods have the potential to address key limitations of previous research.

Weaknesses of the methods: Not the largest or most recent GWASes were used to generate PGSes.

Strengths of the results: The authors included a comprehensive array of analyses.

Weaknesses of the results: Results are only sometimes clearly described and presented.

Appraisal: The authors suggest that their findings provide evidence for policy reforms (e.g., targeting residential environments, family SES, parenting, and schooling).

Impact: Immediate impact is limited given the short follow-up period (2 years), possibly concerns for selection bias and attrition in the data, and some methodological concerns. The authors are transparent about most of these limitations.

---

## [Author Response]

The following is the authors’ response to the previous reviews.

Our comments on the initial eLife assessment

“This study presents a useful inventory of the joint effects of genetic and environmental factors on psychotic-like experiences, and identifies cognitive ability as a potential underlying mediating pathway. The data were analyzed using solid and validated methodology based on a large, multi-center dataset. The claim that these findings are of relevance to psychosis risk and have implications for policy changes are partially supported by the results”

We sincerely appreciate the editor and reviewers for their valuable feedback and their willingness to accommodate our perspectives in the first revision. In this revision, the comments from the reviewers have allowed us to further improve our manuscript. Regarding the eLife assessment, we would like to discuss two points.

Firstly, regarding your point of our “findings are of relevance to psychosis risk…partially supported…”, we want to address that our study is closely related to psychosis risk. Childhood psychotic-like experiences (PLEs) are closely linked to psychotic risk and have been shown to increase the risk of general psychopathology, as mentioned in our Introduction and Discussion.

The reviewers asked for clearer differentiation between PLEs and schizophrenia, which we incorporated in this revision (line 100~111; line 419~430). So, this revised version now clearly points out that findings are relevant primarily to psychosis risk, and only partially relevant to schizophrenia risk.

Secondly, regarding “…implications for policy changes are partially supported…”, we have revised our study’s social contribution more clearly and specifically. Incorporating the comments, we have revised that our study offers an insight to the future studies by showing the importance of integrative approaches, considering multi-factorial neurocognition and psychopathology ranging from genes to environment (line 503~512), rather than offers direct policy implications.

Our collaboration with eLife and the reviewers has proven satisfactory and enriching. The community, coupled with the innovative system and culture established around eLife, has significantly advanced the progression of scientific research. We are privileged to contribute to this endeavor.

Recommendations for the authors:

**Reviewer #1 (Recommendations For The Authors):**
I am happy with the revisions provided by the authors and I think most of my concerns have been addressed satisfactorily. One remaining concern is the authors' conflation of PLEs and schizophrenia. They stated, for example, that it is necessary to adjust for schizophrenia PGS. Even though studies have found a statistical relationship between schizophrenia PGS and PLEs, this relationship is not very strong (although statistically significant) and other studies have found no relationship. Similarly, having PLEs increases the risk of developing psychosis, but that does not necessarily mean that this risk is substantial or specific. I think this needs more nuance in the manuscript and the term 'schizophrenia' should be used sparsely and very carefully as the paper has focused on PLEs. Otherwise, great work on the revisions, thank you.

Thank you for your comment on the use of PLEs and schizophrenia. We clearly understand the differences between the two and we made relevant corrections throughout the manuscript. In particular, we added that PLEs are not a direct predictor of schizophrenia and corrected any expressions that may imply that PLEs are closely related to schizophrenia in the Introduction.

“Psychotic-like experiences (PLEs), which are prevalent in childhood, indicate the risk of psychosis (van der Steen et al., 2019; Van Os & Reininghaus, 2016). Although they are not a direct precursor of schizophrenia, children reporting PLEs in ages of 9-11 years are at higher risk of psychotic disorders in adulthood (Kelleher & Cannon, 2011; Poulton et al., 2000). PLEs also point towards the potential for other psychopathologies including mood, anxiety, and substance disorders (van der Steen et al., 2019), are linked to deficits in cognitive intelligence (Cannon et al., 2002; Kelleher & Cannon, 2011) and show a stronger association with environmental risk factors during childhood than other internalizing/externalizing symptoms (Karcher, Schiffman, et al., 2021).

Maladaptive cognitive intelligence may act as a mediator for the effects of genetic and environmental risks on the manifestation of psychotic symptoms (Cannon et al., 2000; Keefe et al., 2006; Reichenberg et al., 2005).” (line 100~111)

We also revised any expressions that could be perceived as implying relevance to schizophrenia in the Discussion.“Prior research identifying the mediation of cognitive intelligence focused on either genetic (Karcher, Paul, et al., 2021) or environmental factors (Lewis et al., 2020) alone. Studies with older clinical samples have shown that cognitive deficit may be a precursor for the onset of psychotic disorders (Eastvold et al., 2007; Fett et al., 2020; Vorstman et al., 2015). Our study advances this by demonstrating the integrated effects of genetic and environmental factors on PLEs through the cognitive intelligence in 9-11 years old children. Such comprehensive analysis contributes to assessing the relative importance of various factors influencing children's cognition and mental health, and it can aid future studies designed for identifying health policy implications. Considering the directions and magnitudes of the effects, though the effects of PGS remain significant, aggregated effects of environmental factors account for much greater degrees on PLEs.” (line 419~430)

**Reviewer #2 (Recommendations For The Authors):**
I thank the authors for addressing most of my comments. I feel the manuscript has already greatly improved.I have a few more comments.1. Although I did not make this comment, I find the authors' reply to the following comment by Reviewer #1 unclear:Original comment 'I like that the assessment of CP (cognitive performance) and self-reports PLEs is of good quality. However, I was wondering which 4 items from the parent-reported CBCL (Child Behavior Checklist) were used and how did they correlate with the child-reported PLEs? And how was distress taken into account in the child self-reported PLEs measurement? Which PLEs measures were used?'The authors' response refers to correlation coefficients, but I think Reviewer #1's inquiry was on more than these correlations.

Thank you for your concern. We think that this comment was referring to our previous manuscript submitted elsewhere. In our initial submission to eLife, we already added the details about the four items from the parent-reported CBCL and how distress was considered in the child self-reported PLEs measurement (Appendix S1, page 48).

1. Regarding the authors' reply that they have 'standardized the use of 'cognitive capacity' - I do not understand what this means.How exactly was this term standardized? In fact, I can find the term 'cognitive capacity' only once and it seemed to have been deleted from the manuscript. This is fine, but it doesn't clearly align with the statement that this term has been standardized.

We apologize for causing such confusion. What we meant was that throughout our revised manuscript, we used the term “cognitive phenotypes” instead of “cognitive capacity”.

1. Regarding my initial comment that 'it needs to be described how cognitive performance was defined in Lee 2018.' - I believe this is still not clarified.The authors write 'CP was measured as the respondent's score on cognitive ability assessments', but it remains unclear what exactly these assessments were.

Thank you for pointing this out. We added that “CP, measured as the respondent's score on cognitive ability assessments of general cognitive function and verbal-numerical reasoning, was assessed in participants from the COGENT consortium and the UK Biobank” (line 204~206).

1. Regarding the authors' reply to my comment 'In the 'Path Modeling' section, please explain what 'factors and components' concretely refer to. How is this different from a standard SEM with latent factors?'I can see that the authors explained 'components' (=the weighted sum of observed variables), but please also add what you mean by 'factors' - and how these are different from 'components' (line 284).Furthermore, I don't think it is correct that SEMs can only model latent factors, but not components (=measured variables). I also cannot see how using a weighted sum of observed variables controls more effectively for bias in estimation than latent factors. However, even though I do have some knowledge on this method, I'm not an expert and would appreciate the authors, other reviewer and/or editor to weigh in on this point.

Thank you for pointing this out. We added that latent factors are indirectly measured indicators that explain the covariance among observed variables (line 263~271). We also added that standard SEM method using latent factors assumes that observed variables within each construct share a common underlying factor, but if this assumption is not met, then the standard SEM method cannot effectively control for biases. This is the reason why the IGSCA method, which addresses this limitation by allowing for use of both composite and latent factors as constructs.

“Standard SEM using latent factors (i.e., indirectly measured indicators that explain the covariance among observed variables) to represent indicators such as PGS or family SES relies on the assumption that observed variables within each construct share a common underlying factor. If this assumption is violated, standard SEM cannot effectively control for estimation biases. The IGSCA method addresses this limitation by allowing for the use of composite indicators (i.e., components)—defined as a weighted sum of observed variables—as constructs in the model, more effectively controlling bias in estimation compared to the standard SEM. During estimation, the IGSCA determines weights of each observed variable in such a way as to maximize the variances of all endogenous indicators and components.” (line 263~271)

1. I overall disagree with the authors' following statement 'It has been suggested from prior studies that these variables (PGS, family SES, neighborhood SES, positive family and school environment, and PLEs) are less likely to share a common factor', but I appreciate the authors' argument.

Thank you for your comment. To make clarify our statement in the manuscript, we changed the sentence to “Considering that the observed variables of the PGSs, family SES, neighborhood SES, positive family and school environment, and PLEs are evaluated as a composite index by prior research, the IGSCA method can mitigate bias more effectively by representing these constructs as components” (line 274~277).

1. Regarding 'genetic ethnicity': please describe your methods on how this was defined.

Genetic ethnicity was defined as the genetic ancestry of participants, which is included as one of observations in the original ABCD Study data. To avoid further confusion, we corrected ‘genetic ethnicity’ to ‘genetic ancestry’ throughout the manuscript.

1. Regarding 'a more direct genetic predictor of PLEs' - I still don't understand what the contrast is here. More direct than what else?

The description was unclear; we removed it from our manuscript.

1. Regarding the factor loadings in Figure 3: I don't understand how deprivation loads positively on 'low neighborhood SES', but poverty loads negatively. Shouldn't they both show the same direction of effect/loading on neighbourhood SES, while 'years of residency' should show the opposite direction (i.e., deprivation and poverty = risk, while years of residency = protective)? Are these unexpected loadings?The authors did not yet respond to this point: 'Please also add the autocorrelations between the 3 PLE measures. I assume these were also modelled statistically, given the strong correlations between time points?' Were these correlations not modelled? Why not?Figure 3B is still unclear. Was intelligence included here? What is the difference between Figure 3A and B? The legend suggests that 3B shows the indirect effects, but figure 3B looks like a direct effect, while 3A seem to show the indirect effect.

The reviewer’s confusion resulted from our incorrect description. The factor loadings of low neighborhood SES were marked incorrectly. The loading for ‘years of residence’ and ‘poverty’ should be switched: -0.3648 for ‘years of residence’ and +0.877 for ‘poverty’. This was a mistake when we were applying factor loadings in the Figure. We thank you for pointing this out.

We apologize for missing your point on autocorrelation. Adding autocorrelations between the three PLEs is unrelated to our research goal. In this paper, we investigated how genetic and environmental factors explain the variations in PLEs between participants, regardless of changes over time. Since we used PLEs of multiple follow-ups to ensure that the results are robust irrespective of the timing of PLE measurements, taking autocorrelation into account is not necessary.

The decision to add autocorrelation, which involves using the outcome variable at time (t-1) as a predictor for the outcome variable at time t, depends on the research focus. If your interest lies in explaining inter-individual variation in the rate of change in PLEs over a one-year period, then autocorrelation should be controlled for (typically, predictors measured at different time points are used in such cases). However, this was not the focus of this paper, which is why we did not apply autocorrelation in the SEM analysis.

We apologize for the confusion between Figure 3A and 3B. To clarify, we added titles in the figure images as “Direct effects” and “Indirect effects”. We also changed the legend as well.

“A. Direct pathways from PGS, high family SES, low neighborhood SES, and positive environment to cognitive intelligence and PLEs. Standardized path coefficients are indicated on each path as direct effect estimates (significance level *p<0.05). B. Indirect pathways to PLEs via intelligence were significant for polygenic scores, high family SES, low neighborhood SES, and positive environment, indicating the significant mediating role of intelligence.” (line 968~973)

Figure 3A shows direct effects: i.e., the coefficients of paths from PGS, family SES, neighborhood SES, and positive environment to intelligence and PLEs, as well as the coefficient of paths from intelligence to PLEs. This is why Figure 3A shows colored arrows starting from PGS, family and neighborhood SES, and positive environment towards intelligence and PLEs, as well as the arrows from intelligence to PLEs. On the other hand, in Figure 3B, the colored arrows staring from PGS, family and neighborhood SES, and positive environment goes through intelligence, and heads towards PLEs. This was meant to show that the indirect effects shown in Figure 3B indicate the specific effects of PGS, family SES, neighborhood SES, and positive environment on PLEs mediated by intelligence.

In short, Figure 3 can be seen as a diagram drawn from Table 2: direct effects of the genetic and environmental variables on intelligence and PLEs, and direct effects of intelligence on PLEs are shown in Figure 3A; indirect effects of genetic and environmental variables on PLEs mediated by intelligence are shown in Figure 3B.

1. Regarding Supporting Information tables: to make these more digestible, I suggest using Excel and adding one table per sheet with a clear title and legend, indicating what each table shows.For example, Table S1 has 9(?) different subsections, all called the same (Linear Mixed Model: Multiethnic). It is not clear how each subsection differs from the others. Separate tables in separate excel sheets might be easier.Also, I think two decimal points might be good enough, enhancing readability of these tables.

Thank you for your suggestion. We moved the supplementary tables into an external Excel file, with each sheet showing different tables, as well as titles, legends, and clear subsections.

1. Regarding reporting exact p-values in Table 2: I don't understand. At the moment, categorical significance statements are reported. Were these not based on exact p-values (or how else was it decided if a finding was significant at a 0.05 (?) significance level).

Either remove the significance column completely (as p-values cannot be estimated due to non-normality) or specify exactly/clarify what this column shows and this was derived.

We apologize for the confusion. In Table 2, we checked the significance of each path using 95% confidence intervals with 5,000 bootstrapping iterations. Since 95% confidence intervals that does not include zero is equivalent to p-values below 0.05 significance level, we believe this is an appropriate alternative for reporting the significance of each path in the SEM model.

We specified the reason why we were not able to calculate exact p-values (clean copy: line 299~303). “As a trade-off for obtaining robust nonparametric estimates without distributional assumptions for normality, the IGSCA method does not return exact p-values (Hwang, Cho, Jung, et al., 2021). As a reasonable alternative, we obtained 95% confidence intervals based on 5,000 bootstrap samples to test the statistical significance of parameter estimates.”